# Learning a Neural-network-based Representation for Open Set Recognition

## Abstract

In this paper, we present a neural network based representation for addressing the open set recognition problem. In this representation instances from the same class are close to each other while instances from different classes are further apart, resulting in statistically significant improvement when compared to other approaches on three datasets from two different domains.

## 1 Introduction

To build robust AI systems, Dietterich Dietterich (2017) reasons that one of the main challenges is handling the "unknown unknowns." One idea is to detect model failures (i.e., the system understands that its model about the world/domain has limitations and may fail). For example, assume you trained a binary classifier model to discriminate between pictures of cats and dogs. Let's assume this model performs well at recognizing images of cats and dogs. What would this model do if it is faced with a picture of a fox or a caracal? The model being a binary classifier will predict these pictures to be either a dog or a cat, which is not desirable and can be considered as a failure of the model. In machine learning, one direction of research for detecting model failure is "open category learning", where not all categories are known during training, and the system needs to appropriately handle instances from novel/unknown categories that may appear during testing. Besides "open category learning", terms such as "open world recognition" Bendale & Boult (2015) and "open set recognition" Scheirer et al. (2013); Bendale & Boult (2016) have been used in past literatures. In this paper, we will use the term "open set recognition".

Where does open set recognition appear in real-world problems? Various real-world applications operate in an open set scenario. For example, Ortiz and Becker Ortiz & Becker (2014) point to the problem of face recognition. One such use case is automatic labeling of friends in social media posts, "where the system must determine if the query face exists in the known gallery, and, if so, the most probable identity." Another domain is in malware classification, where training data usually is incomplete because of novel malware families/classes that emerge regularly. As a result, malware classification systems operate in an open set scenario.

In this paper, we propose a neural network based representation and a mechanism that utilizes this representation for performing open set recognition. Since our primary motivation when developing this approach was the malware classification domain, we evaluate our work on two malware datasets. To show the applicability of our approach to other domains, we evaluate our approach on images.

Our contributions include: (1) we propose an approach for learning a representation that facilitates open set recognition, (2) we propose a loss function that enables us to use the same distance function both when training and when computing an outlier score, (3) our proposed approaches achieve statistically significant improvement compared to previous research work on three datasets.

## 2 Related Work

We can broadly categorize existing open set recognition systems into two types. The first type provides mechanisms to discriminate known class instances from unknown class instances. These systems, however, cannot discriminate between the known classes, where there is more than one. Research works such as Scheirer et al. (2013); Bodesheim et al. (2013; 2015) fall in this category. Scheirer et al. Scheirer et al. (2013) formalized the concept of open set recognition and proposed a

1-vs-set binary SVM based approach. Bodesheim et al. Bodesheim et al. (2013) propose KNFST for performing open set recognition for multiple known classes at the same time. The idea of KNFST is further extended in Bodesheim et al. (2015) by considering the locality of a sample when calculating its outlier score.

The second type of open set recognition system provides the ability to discriminate between known classes in addition to identifying unknown class instances. Research works such as Jain et al. (2014); Bendale & Boult (2015; 2016); Da et al. (2014); Ge et al. (2017) fall in this category. PI-SVM Jain et al. (2014), for instance, uses a collection of binary SVM classifiers, one for each class, and fits a Weibull distribution over the score of each classifier. This approach allows PI-SVM to be able to both perform recognition of unknown class instances and classification between the known class instances. Bendale and Boult Bendale & Boult (2015) propose an approach to extend Nearest Class Mean (NCM) to perform open set recognition with the added benefit of being able to do incremental learning.

Neural Net based methods for open set recognition have been proposed in Bendale & Boult (2016); Cardoso et al. (2015); Ge et al. (2017). Openmax Bendale & Boult (2016) (a state-of-art algorithm) modifies the regular Softmax layer of a neural network by redistributing the activation vector (the values of the final layer of a neural network that are given as input to the Softmax function) to account for unknown classes. Ge et al. Ge et al. (2017) use DCGAN Radford et al. (2015) to generate unknown-class samples. The network is trained on the original instances plus the generated samples, and Openmax is used to ajdusted the activation vector. Similarly, Yu et al. Yu et al. (2017) generate "negative" samples using adversarial learning and use supervised algorithms to learn the final classifier. Our approach does not generate "unknown-class/negative" samples and hence does not increase the training overhead in the learning step. In malware classification, K. Rieck et al.Rieck et al. (2011) proposed a malware clustering approach and an associated outlier score. Although the authors did not propose their work for open set recognition, their outlier score can be used for unsupervised open set recognition. Rudd et al. Rudd et al. (2017) outline ways to extend existing closed set intrusion detection approaches for open set scenarios. Lee et al. Lee et al. (2018) are interested in detecting test samples that are out-of-distribution (different from the training "in-distribution'); however, each test sample is still from one of the known classes.

## 3 APPROACH

For open set recognition, given a set of instances belonging to known classes, we would like to learn a function that can accurately classify an unseen instance to one of the known classes or an unknown class. Let $D$ be a set of instances $X$ and their respective class labels $Y$ (i.e., $D = (X, Y)$), and $K$ be the number of unique known class labels. Given $D$ for training, the problem of open set recognition is to learn a function $f$ that can accurately classify an unseen instance (not in $X$) to one of the $K$ classes or an unknown class (or the "none of the above" class).

The problem of open set recognition differs from the problem of closed set ("regular") classification because the learned function $f$ needs to handle unseen instances that might belong to classes that are not known during training. That is, the learner is robust in handling instances of classes that are not known. This difference is the main challenge for open set recognition. Another challenge is how to learn a more effective instance representation that facilitates open set recognition than the original instance representation used in $X$.

**Learning representations** Consider $\vec{x}$ is an instance and $y = f(\vec{x})$ is the class label predicted using $f(\vec{x})$. In case of a closed set, $y$ is one of the known class labels. In the case of open set, $y$ could be one of the known classes or an unknown class. The hidden layers in a neural network, $\vec{z} = g(\vec{x})$, can be considered as different representations of $\vec{x}$. Note, we can rewrite $y$ in terms of the hidden layer as $y = f(\vec{z}) = f(g(\vec{x}))$.

The objective of our approach is to learn a representation that facilitates open set recognition. We would like this new representation to have two properties: (P1) instances of the same class are closer together, and (P2) instances of different classes are further apart. The two properties can lead to larger spaces among known classes for the instances of the unknown classes to occupy. Consequently, instances of unknown classes could be more effectively detected. This representation is similar in spirit to a Fisher Discriminant. A Fisher discriminant aims to find a linear projection that maximizes

---

**Algorithm 1:** Training to minimize ii-loss.

---

**Input** :
      $(X, Y)$: Training data and labels

1 **for** *number of training iterations* **do**
2     Sample a mini-batch $(X_{batch}, Y_{batch})$ from $(X, Y)$
3     $Z_{batch} \leftarrow g(X_{batch})$
4     $\{\vec{\mu}_1 \cdots \vec{\mu}_K\} \leftarrow class\_means(Z_{batch}, Y_{batch})$
5     ii-loss $\leftarrow intra\_spread(Z_{batch}, \{\vec{\mu}_1 \cdots \vec{\mu}_K\})$ - $inter\_separation(\{\vec{\mu}_1 \cdots \vec{\mu}_K\})$
6     update parameters of $g$ using stochastic gradient descent to minimize ii-loss

7 $\{\vec{\mu}_1 \cdots \vec{\mu}_K\} \leftarrow class\_means(g(X), Y)$
8 return $\{\vec{\mu}_1 \cdots \vec{\mu}_K\}$ and parameters of $g$ as the model.

---

between class (inter-class) separation while minimizing within class (intra-class) spread. Such a projection is obtained by maximizing the Fisher criteria. However, in the case of this work, we use a neural network with a *non-linear* projection to learn this representation. The neural network $g$ used to learn the representation can be either a combination of convolution and fully connected layers, as shown in Figure 1a, or it can be all fully connected layers, Figure 1b. Both types are used in our experimental evaluation.

**II-Loss Function**     In a typical neural network classifier, the activation vector that comes from the final linear layer is given as input to a Softmax function. Then the network is trained to minimize a loss function such as cross-entropy on the outputs of the Softmax layer. In our case, the output vector $\vec{z}_i$ of the final linear layer of a neural network (i.e., activation vector that serves as input to a softmax in a typical neural net) are considered as the projection of the input vector $\vec{x}_i$, of instance $i$, to a different space. The network is trained using mini-batch stochastic gradient descent with backpropagation as outlined in Algorithm 1 to minimize the loss function in Equation 1, which we will refer to ii-loss for the remainder of this paper. In this loss function, we aim to maximize the distance between different classes (inter-class separation) and minimize the distance of an instance from its class mean (intra-class spread). We measure intra-class spread as the average distance of instances from their class means (first part of Equation 1). We measure the inter-class separation in terms of the distance between the closest two class means among all the $K$ known classes (second part of Equation 1). After the network finishes training, the class means are calculated for each class using all the training instances of that class and stored as part of the model.

$$ii\text{-}loss = \Big( \underbrace{\frac{1}{N} \sum_{j=1}^{K} \sum_{i=1}^{|C_j|} \|\vec{\mu_j} - \vec{z_i}\|_2^2}_{\text{intra\_spread}} \Big) - \Big( \underbrace{\min_{\substack{1 \leq m \leq K \\ m+1 \leq n \leq K}} \|\vec{\mu_m} - \vec{\mu_n}\|_2^2}_{\text{inter\_sparation}} \Big) \tag{1}$$

where $|C_j|$ is the number of training instances in class $C_j$, $N$ is the number of training instances, $K$ is the number of known classes, and $\vec{\mu_j} = \frac{1}{|C_j|} \sum_{i=1}^{|C_j|} \vec{z_i}$ is the mean of class $C_j$.

**Combining ii-loss with Cross Entropy Loss**     While the two desirable properties P1 and P2 discussed in an earlier Section aim to have a representation that separates instances from different classes, lower classification error is not explicitly stated. Hence, a third desirable property (P3) is a low classification error in the training data. To achieve this, alternatively, a network can be trained on both cross entropy loss and ii-loss (Eq 1) simultaneously. The network architecture in Figure 1c can be used. In this configuration, an additional linear layer is added after the z-layer. The output of this linear layer is passed through a Softmax function to produce a distribution over the known classes. Although Figure 1c shows a network with convolutional and fully connected layers, combining ii-loss with cross-entropy can also work with a network of fully connected layers only. The network is trained using mini-batch stochastic gradient descent with backpropagation. During each training iteration, the network weights are first updated to minimize on ii-loss and then in a separate step updated to minimize cross entropy loss. Other researchers have trained neural networks using more

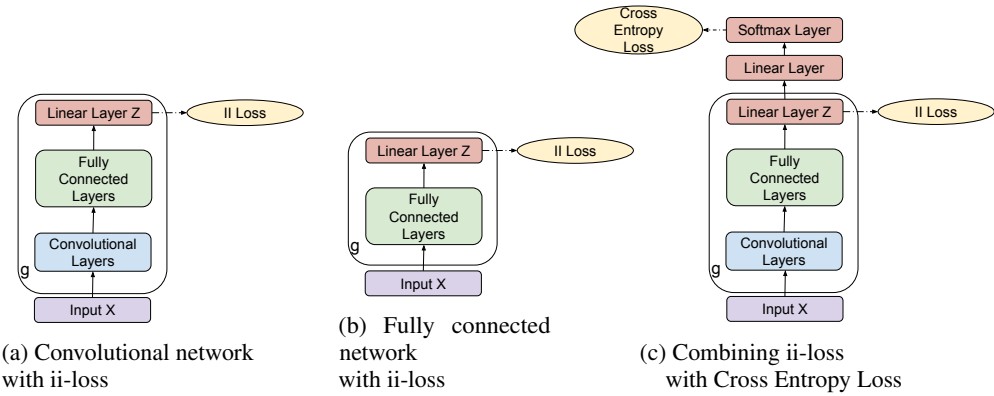

(a) Convolutional network with ii-loss

(b) Fully connected network with ii-loss

(c) Combining ii-loss with Cross Entropy Loss

Figure 1: Network architecture with ii-loss.

than one loss function. For example, the encoder network of an Adversarial autoencoder Makhzani et al. (2015) is updated both to minimize the reconstruction loss and the generators loss.

**Outlier Score for Open Set Recognition**  During testing, we use an outlier score to indicate the degree to which the network predicts an instance $\vec{x}$ to be an outlier. This outlier score is calculated as the distance of an instance to the closest class mean from among $K$ known classes.

$$outlier\_score(\vec{x}) = \min_{1 \leq j \leq K} \|\vec{\mu}_j - \vec{z}\|_2^2 \qquad (2)$$

where $\vec{z} = g(\vec{x})$. Because the network is trained to project the members of a class as close to the class mean as possible the further away the projection $\vec{z}$ of instance $\vec{x}$ is from the closest class mean, the more likely the instance is an outlier for that class.

**Threshold Estimation**  Once an outlier score identified, the next step is determining what threshold value of this score will indicate an outlier. In other words, how far does the projection of an instance need to be from the closest class mean for it to be deemed an outlier. For this work, we propose a simple threshold estimation. To pick an outlier threshold, we assume that a certain percent of the training set to be noise/outliers . We refer to this percentage as the contamination ratio. For example, if we set the contamination ratio to be 0.01, it will be like assuming 1% of the training data to be noise/outliers. Then, we calculate the outlier score on the training set instances, sort the scores in ascending order and pick the 99 percentile outlier score value as the outlier threshold value. The reader might notice that the threshold proposed in this section is a global threshold. This means that the same outlier threshold value is used for all classes. An alternative to this approach is to estimate the outlier threshold per-class. However, in our evaluation, we observe that global threshold consistently gives more accurate results than the per-class threshold.

**Performing Open Set Recognition**  Open set recognition is a classification over $K + 1$ class labels, where the first $K$ labels are from the known classes the classifier is trained on, and the $K + 1$st label represents the *unknown* class. This is performed using the outlier score in Equation 2 and the associated *threshold*. The *outlier_score* of a test instance is first calculated. If the score is greater than *threshold*, the test instance is labeled as $K + 1$, which in our case corresponds to the *unknown class*; otherwise, the appropriate class label is assigned to the instance from among the known classes, Equation 3. The predicted class probability over the known classes, we can be expressed as the softmax of the negative distance of a projection $\vec{z}$, of the test instance $\vec{x}$ (i.e., $\vec{z} = g(\vec{x})$), from all the known class means, Equation 4. Note, when a network network trained on both ii-loss and cross entropy loss $P(y = k \mid \vec{x})$ is from the Softmax layer in Figure 1c.

$$y = \begin{cases} K + 1, & \text{if } outlier\_score > threshold \\ \underset{1 \leq j \leq K}{\operatorname{argmax}} P(y = j \mid \vec{x}), & \text{otherwise} \end{cases} \qquad (3)$$

$$P(y = j \mid \vec{x}) = \frac{e^{-\|\vec{\mu}_j - \vec{z}\|_2^2}}{\sum_{m=1}^{K} e^{-\|\vec{\mu}_m - \vec{z}\|_2^2}} \qquad (4)$$

## 4 EVALUATION

**Datasets and Simulating Open Set Dataset**   We evaluate our approach using three datasets. The first is the Microsoft Malware Challenge Dataset msc (2015) which consists of disassembled windows malware samples from 9 malware families/classes. We use 10260 samples which are disassembled file parser was able to process correctly. The second dataset is the Android Genome Project Dataset mal which consists of malicious Android apps. In our evaluation, we use only 9 classes that have at least 40 samples. After removing the smaller classes, the dataset has 986 samples. We extract function call graph (FCG) features from the malware samples as proposed by Hassen and Chan Hassen & Chan (2017) . In case of the Android samples Android dataset we first use ada to extract the functions and the function instructions and then used Hassen & Chan (2017) to extract the FCG features. For MS Challenge dataset, we reformat the FCG features as a graph adjacency matrix by taking the edge frequency features in Hassen & Chan (2017) and rearranging them to form an adjacency matrix. Formatting the features this way allowed us to use convolutional layers on the MS Challenge dataset. To show that our approach can be applied to other domains we also evaluate our work on the MNIST Datasetmni, which consists of images of handwritten digits from 0 to 9.

To simulate an open world dataset for our evaluation datasets, we randomly choose $K$ number of classes from the dataset, which we will refer to as known classes in the remainder of this evaluation section, and keep only training instances from these classes in the training set. We will refer to the other classes as unknown classes. In case of the MS Dataset and Android Dataset, first, we randomly chose 6 known classes and treat set the remaining 3 as unknown classes. We then randomly select 75% of the instances from the known classes for the training set and the remaining for the test set. We further withhold one-third of the test set to serve as a validation set for hyperparameter tuning. We use only the known class instances for tuning. In these two datasets, all the unknown class instances are placed into the test set. In case of the MNIST dataset, first, we randomly chose 6 known classes and the remaining 4 as unknown classes. We then remove the unknown class instances from the training set. We leave the test set, which has both known and unknown class instances, as it is. For each of our evaluation datasets, we create 3 open set datasets. We will refer to these open set datasets as OpenMNIST1, OpenMNIST2, and OpenMNIST3 for the three open set evaluation datasets created from MNIST. Similarly, we also create OpenMS1, OpenMS2, and OpenMS3 for MS Challenge dataset and OPenAndroid1, OpenAndroid2, and OpenAndroid3 for Android Genom Project dataset.

**Evaluated Approaches**   We evaluate five approaches; all implemented using Tensorflow. The first (*ii*) is a network setup to be trained using ii-loss. The second (*ii+ce*) is a network setup to be simultaneously trained using ii-loss and cross entropy (Section 8). The third (*ce*) is a network which we use to represent the baseline, is trained using cross-entropy only (network setup in Figure 1c without the ii-loss.) The forth approach is OpenmaxBendale & Boult (2016) (a state-of-art algorithm), which was reimplemented based on the original paper and the authors' source code to fit our evaluation framework. The authors of Openmax state that the choice of distance function Euclidean or combined Euclidean and cosine distance give similar performance in the case of their evaluation datasets Bendale & Boult (2016). In our experiments, however, we observed that the combined Euclidean and cosine distance gives a much better performance. So we report the better result from combined Euclidean and cosine distance. The final approach is Generative Openmax (G-Openmax) Ge et al. (2017). The networks used for MS and MNIST datasets have convolutional layers at the beginning followed by fully connected layers, whereas for the android dataset we use only fully connected layers. The architecture is detailed Appendix A. Our source code is available on Github [1]. The evaluation datasets are available online on their respective websites.

### 4.1 DETECTING UNKNOWN CLASS INSTANCES AND OPEN SET RECOGNITION

We start our evaluation by showing how well $outlier\_score$ (in Equation 2) is able to identify unknown class instances. We evaluate it using 3 random open set datasets created from MS, Android

---

[1]https://github.com/shrtCKT/opennet

Table 1: Average AUC of 30 runs up to 100% FPR and 10% FPR (the positive label represented instances from unknown classes and the negative label represented instances from the known classes when calculating the AUC). The underlined average AUC values are higher with statistical significance (p-value < 0.05 with a t-test) compared to the values that are not underlined on the same row. The average AUC values in **bold** are the largest average AUC values in each row.

|  | FPR | ce | ii | ii+ce |
|---|---|---|---|---|
| MNIST | 100% | 0.9282 ($\pm$0.0179) | **0.9588** ($\pm$0.0140) | 0.9475 ($\pm$0.0151) |
|  | 10% | 0.0775 ($\pm$0.0044) | **0.0830** ($\pm$0.0045) | 0.0801 ($\pm$0.0044) |
| MS Challenge | 100% | 0.9143 ($\pm$0.0433) | 0.9387 ($\pm$0.0083) | **0.9407** ($\pm$0.0135) |
|  | 10% | 0.0526 ($\pm$0.0091) | **0.0623** ($\pm$0.0030) | 0.0596 ($\pm$0.0035) |
| Android Genom | 100% | 0.7755 ($\pm$0.1114) | 0.8563 ($\pm$0.0941) | **0.9007** ($\pm$0.0426) |
|  | 10% | 0.0066 ($\pm$0.0052) | 0.0300 ($\pm$0.0193) | **0.0326** ($\pm$0.0182) |

Table 2: Average F-Score of 30 Runs. The underlined average AUC values are higher with statistical significance (p-value < 0.05 with a t-test) compared to the values that are not underlined on the same row. The average AUC values in **bold** are the largest average AUC values in each row.

|  | Openmax | G-Openmax | ce | ceii | ii |
|---|---|---|---|---|---|
| MNIST | 0.88($\pm$0.05) | 0.69($\pm$0.02) | 0.74($\pm$0.20) | 0.92($\pm$0.02) | **0.93**($\pm$0.02) |
| MS | 0.87($\pm$0.01) | 0.83($\pm$0.02) | 0.86($\pm$0.04) | **0.89**($\pm$0.01) | 0.88($\pm$0.01) |
| Android | 0.30($\pm$0.12) | 0.60($\pm$0.11) | 0.46($\pm$0.10) | **0.71**($\pm$0.17) | 0.69($\pm$0.15) |

and MNIST datasets as discussed in the Section on simulating open set dataset. For example, in the case of MNIST dataset, we run 10 experiments on OpenMNIST1, 10 experiments on OpenMNIST2, and 10 experiments on OpenMNIST3. We then report the average of the 30 runs. We do the same for the other two datasets.

Table 1 shows the results of this evaluation. To report the results in such a way that is independent of outlier threshold, we report the area under ROC curve (AUC). This area is calculated using the outlier score and computing the true positive rate (TPR) and the false positive rate (FPR) at different thresholds. We use the t-test to measure the statistical significance of the difference in AUC values. Looking at the AUC up to 100% FPR in all three datasets, our approach *ii* and *ii+ce* perform significantly better(with p-value of 0.04 or less) in identifying unknown class instances than the baseline approach *ce* (using only cross entropy loss.) Although AUC up to 100% FPR gives a full picture, in practice it is desirable to have good performance at lower false positive rates. That is is why we report AUC up to 10% FPR. Our two approaches report a significantly better AUC than the baseline network trained to only minimize cross entropy loss. We didn't include Openmax in this section's evaluation because it doesn't have an explicit outlier score.

When the proposed approach is used for open set recognition, the final prediction is a class label, which can be one of the $K$ known class labels if the test instances has an outlier score less than a threshold value or it can be an "*unknown*" label if the instance has an outlier score greater than the threshold, Eq. 3. In addition to the three approaches evaluated in the previous section, we also include Openmax Bendale & Boult (2016) and G-Openmax Ge et al. (2017) in these evaluations because they give final class label predictions.

We use average F-score to evaluate open set recognition performance and t-test for statistical significance. Using the same experimental setup the earlier experiment, we report the result of the average f-score, averaged across all class labels and across 30 experiment runs in Table 2. On all three datasets the *ii* and *ii+ce* networks gives significantly better f-score compared to the other two configurations (with p-value of 0.0002 or less). In case of the Android dataset, all networks perform lower compared to the other two datasets. We attribute this to the small number of samples in the Android datasets. The dataset is also imbalanced with many classes only having less than 60 samples.

Two limitations of Openmax can explain its weaker performance compared to our proposed approaches: 1) it does not use a loss function that directly incentivizes projecting class instances around

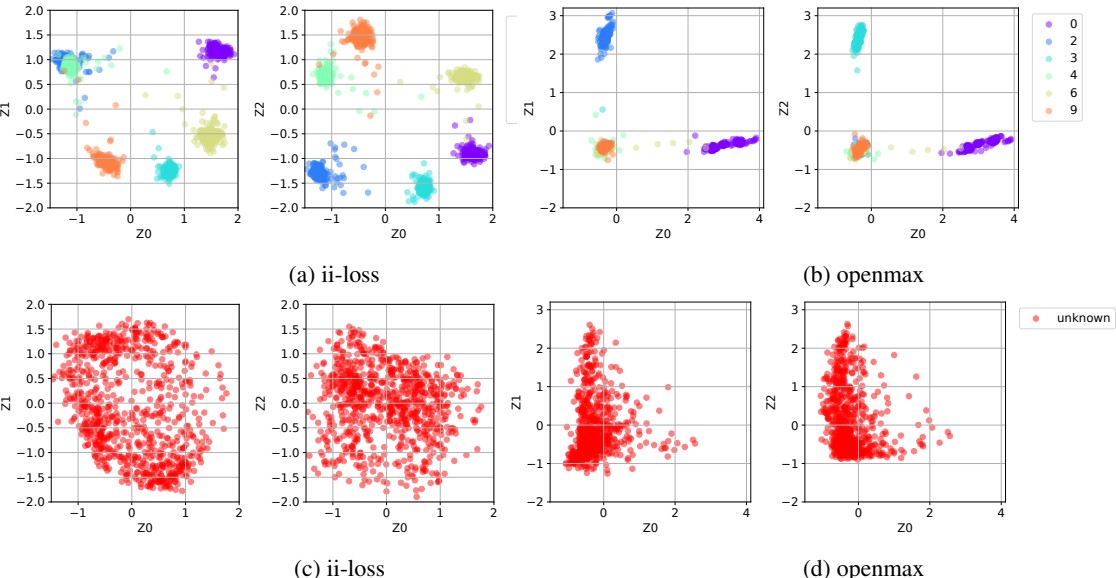

Figure 2: The z-layer projection of (a, b) known and (c, d) unknown class instances from test set of MNIST dataset. The labels 0,2,3,4,6,9 represent the known classes while the label "unknown" represents the unknown classes.

the mean class activation vector and 2) the distance function used by Openmax is not necessarily the right distance function for final activation vector space since it is not used in training. We addressed these limitations by training a neural network with a loss function that explicitly encourages properties P1 and P2. Also, we use the same distance function during training and test.

### 4.2 DISCUSSIONS

Figure 2 provides evidence on how our network projects unknown class instances in the space between the known classes. In the figure the z-layer projection of 2000 random test instances of an open set dataset created from MNIST with 6 known and 4 unknown classes. The class labels 0, 2, 3, 4, 6, and 9 in the figure represent the 6 known classes while the "unknown" label represents all the unknown classes. The network with ii-loss is set up to have a z-layer dimension of 6, and the figure shows a 2D plot of dimension (z0,z1), (z0,z2). The Openmax network also has a similar network architecture and last layer dimension of 6. In case of ii-loss based projection, the instances from the known classes (Figures 2a) are projected close to their respective class while the unknown class instances (Figures 2c) are projected, for the most part, in the region between the classes. In case of Openmax, Figures 2b and 2d, the unknown class instances do not fully occupy the open space between the known classes. In Openmax, most instances are projected along the axis; this is because of the one-hot encoding induced by cross-entropy loss. So compared to Openmax, ii-loss appears to better utilize space "among" the classes.

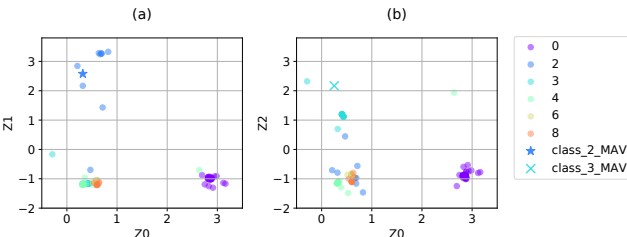

Figure 3: Projections of Android dataset known class test instances from final activation layer of Openmax.

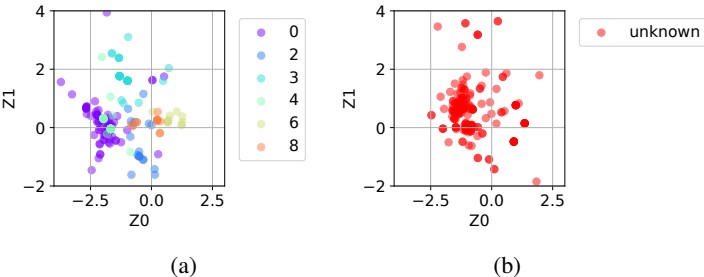

(a)                      (b)

Figure 4: Projections of Android dataset (a) known class and (b) unknown class test instances from z-layer of a network trained with only cross entropy.

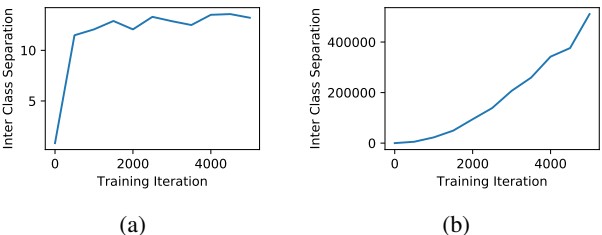

(a)                      (b)

Figure 5: Inter class separation for networks trained (a) with batch normalization used in all layers and (b) without batch normalization at the z-layer.

Performance of Openmax is especially low in case of the Android dataset because of low recall on known classes with small number training instances. The low recall was caused by test instances from the smaller classes being projected further away from the class's mean activation vector (MAV). For example, in Figure 3a we see that test instances of class 2 are further away from the MAV of class 2 (marked by '⋆'). As a result, these test instances are predicted as unknown. Similarly, in Figure 3b instances of class 3 are far away from the MAV of class 3(marked by 'X'). Performance of network trained with only cross entropy (ce) is also low for Android dataset because unknown class instances were projected close to the known classes (Figure 4). As a result, these instances get labeled as known classes. In turn, resulting in a lower precision score for the known classes.

In our experiments, we have observed batch normalization Ioffe & Szegedy (2015) to be extremely important when using ii-loss. Because batch normalization fixes the mean and variance of a layer, it bounds the output of our z-layer in a certain hypercube, in turn preventing the $inter\_separation$ term in ii-loss from increasing indefinitely. This is evident in Figures 5a and 5b. Figure 5a shows the $inter\_separation$ of the network where batch normalization used in all layers including the z-layer; here, the $inter\_separation$ increases in the beginning but levels off. Whereas when batch normalization is not used in the z-layer the $inter\_separation$ term keeps on increasing as seen in Figure 5b; as a result, ii-loss would not converge.

Autoencoders can also be considered as another way to learn a representation. However, autoencoders do not try to achieve properties P1 and P2. One of the reasons is autoencoder training is unsupervised. Another reason is that non-regularized autoencoders fracture the manifold into different domains resulting in the representation of instances from the same class being further apart Makhzani et al. (2015). Therefore, in the learned representation, the known classes are not well separated. Additionally, outliers get projected to roughly the same area as the known classes. A figure in Appendix C shows the output of an encoder in an autoencoder.

## 5   CONCLUSION

We presented an approach for learning a neural network based representation that projects instances of the same class closer together while projecting instances of the different classes further apart. Our empirical evaluation shows that the two properties lead to larger spaces among classes for instances

of unknown classes to occupy, hence facilitating open set recognition. We compared our proposed approach with a baseline network trained to minimize a cross entropy loss and with Openmax (a state-of-art neural network based open set recognition approach). We evaluated the approaches on datasets of malware samples and images and observed that our proposed approach achieves statistically significant improvement. We proposed a simple threshold estimation technique in this paper. However, there is room to explore a more robust way to estimate the threshold. We leave this for future work.

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

## A  EVALUATION NETWORK ARCHITECTURES

We evaluate 5 networks: ii, ce, ii+ce, Openmax and G-Openmax. The first four networks and the final classifier of G-Openmax have the same architecture up to the fully connected z-layer. In case of the MNIST dataset, the input images are of size (28,28) and are padded to get an input layer size (32,32) with 1 channel. Following the input, layer are 2 non-linear convolutional layers with 32 and 64 units (filters) which have a kernel size of (4,4) with a (1,1) strides and SAME padding. The network also has max polling layers with a kernel size of (3,3), strides of (2,2), and SAME padding after each convolutional layer. Two fully connected non-linear layers with 256 and 128 units follow the second max pooling layer. Then a linear z-layer with a dimension of 6 follows the fully connected layers. In the case of ii+ce and ce networks, the output of the z-layer is fed to an additional linear layer of dimension 6 which is then given to a softmax function. We use Relu activation function for all the non-linear layers. Batch normalization is used throughout all the layers. We also use Dropout with keep probability of 0.2 for the fully connected layers. Adam optimizer with a learning rate of 0.001, beta1 of 0.5, and beta2 of 0.999 is used to train our networks for 5000 iterations. In case of the Openmax network, the output of the z-layer is directly fed to a softmax layer. Similar to the Openmax paper we use a distance that is a weighted combination of normalized Euclidean and cosine distances. For the ce, ii, and ii+ce we use contamination ratio of 0.01 for the threshold selection.

The open set experiments for MS Challenge dataset also used similar architectures as the four networks used for MNIST dataset with the following differences. The input layer size MS Challenge dataset is (67,67) with 1 channel after padding the original input of (63,63). Instead of the two fully connected non-linear layers, we use one fully connected layer with 256 units. We use dropout in the fully connected layer with keep probability of 0.9. Finally, the network was trained using Adam optimizer with 0.001 learning rate, 0.9 beta1, and 0.999 beta2.

We do not use a convolutional network for the Android dataset open set experiments. We use a network with one fully connected layer of 64 units. This is followed by a z-layer with a dimension of 6. For ii+ce and ce networks we further add a linear layer with a dimension of 6 and the output of this layer is fed to a softmax layer. In case of Openmax, the output of the z-layer is directly fed to the softmax layer. For Openmax we use a distance that is a weighted combination of normalized Euclidean and cosine distances. We use Relu activation function for all the nonlinear layers. We used batch normalization for all layers. We also used Dropout with keep probability of 0.9 for the fully connected layers. We used Adam optimizer with a learning rate of 0.1 and first momentum of 0.9 to train our networks for 10000 iterations. For the ce, ii, and ii+ce we use contamination ratio of 0.01 for the threshold selection.

The closed set experiments use the same set up as the open set experiments with the only difference coming from the dimension of the z-layer. For the MNIST dataset, we used z dimension of 10. For the MS and Android datasets, we use z dimension of 9.

## B    CLOSED SET CLASSIFICATION

In this section, we would like to show that on a closed dataset, a network trained using ii-loss performs comparably to the same network trained using cross entropy loss. For closed set classification, all the classes in the dataset are used for both training and test. For MS and Android datasets, we randomly divide the datasets into training, validation, and test and report the results on the test set. The MNIST dataset is already divided into training, validation, and test.

On closed MNIST dataset, a network trained with cross-entropy achieved a 10-run average classification accuracy of 99.42%. The same network trained using ii-loss achieved an average accuracy of 99.31%. The network trained only on cross-entropy gives better performance than the network trained on ii-loss. The results from a network trained both ii-loss cross entropy loss to achieve an average classification accuracy of 99.40%. This result makes it comparable to the performance of the same network trained using cross-entropy only (with a p-value of 0.22). We acknowledge that both results are not state-of-art as we are using simple network architectures. The primary goal of these experiments is to show that the ii-loss trained network can give comparable results to a cross entropy trained network. On the Android dataset, the network trained on a cross entropy gets an average classification accuracy of 93.10% while ii-loss records 92.68%, but the difference is not significant (with a p-value at 0.43).

## C    DISCUSSION

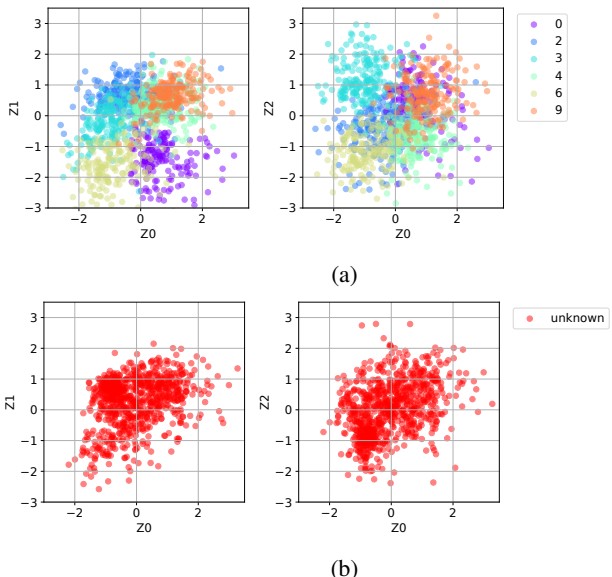

(a)

(b)

Figure 6: Projections of (a) known and (b) unknown class instances using the hidden layer of an Autoencoder. The labels 0,2,3,4,6,9 represent the known classes while the label "unknown" represents the unknown classes.

We mentioned earlier that we used function call graph (FCG) feature for the malware dataset. We also mentioned that in case of the MS Challenge dataset we reformatted the FCG features proposed in Hassen & Chan (2017) to form a $(63, 63)$ adjacency matrix representation of the graph. We feed this matrix as an input to the convolutional network with a (4,4) kernel. Such kernel shape makes sense when it comes to image input because in images proximity of pixels hold essential information. However, it is not apparent to us how nearby cells in a graph adjacency matrix hold meaning full information. We tried different kernel shapes, for example taking an entire row of the matrix at once (because a row of the matrix represents single nodes outgoing edge weights). However the simple (4,4) gives a better close set performance.

