# OpenReview forum: "Learning a Neural-network-based Representation for Open Set Recognition"
_ICLR.cc/2019/Conference_

### Official Review · AnonReviewer1 · 2018-10-17
**The method needs more discussions and more comparison baselines.**

**Rating:** 5
**Confidence:** 4

**Review:**

This paper deals with the open set classification problem, where in addition to the known classes, the method should also be able to recognize the unknown class. The main idea is based on two parts: learning a discriminative representation, and a threshold based detection rule. To learn the embedding, the authors propose to minimize the inner class distance (between each instance to its center) and enlarge the distance between centers. The outlier score of an instance is computed as the minimum distance between known class prototypes. Experiments on various datasets show the ability of the learned method.

I'm not completely sure whether the whole approach is novel or not in the open set recognition domain, but both parts are not novel enough. Pulling similar instances together and pushing dissimilar ones away is the main idea in embedding learning. The ii-loss is similar to the triplet-center loss in the paper "He et al. Triplet-Center Loss for Multi-View 3D Object Retrieval. CVPR18".

Although in the experiments the proposed method achieves good results in most cases, the reviewer suggests the authors comparing with more baselines to make the work solid.
1. Comparing with other embedding learning methods with the same outlier detection score.
The authors should prove that the proposed embedding is important enough in the open set case. For example, using the center loss (Wen et al. A discriminative feature learning approach for deep face recognition. ECCV16), triplet-center loss, triplet loss (computing class centers after embedding).

2. Discuss more on the outlier score part.
How to differentiate the known class outlier and new class? Will the problem be more difficult when the unknown class contains more heterogeneous classes? The authors can also apply existing open set recognition rule on the learned embedding.

Some detailed questions:
1. What's the difference between "the network weights are first updated to minimize on ii-loss and then in a separate step updated to minimize cross entropy loss" and optimize both loss terms simultaneously?
2. "We assume that a certain percent of the training set to be noise/outliers", how to determine the concrete value? Is 1% the helpful one for all cases?
3. Since there is not optimize over the unknown classes in training, could the reason for "the unknown class instances fully occupy the open space between the known classes" is the unknown classes are randomly sampled from the whole class set? For example, if classes about animals are known classes and classes about scene compose the unknown class, will the unknown class also occupy the whole space in this case?
4. What is the motivation of making "the unknown class instances fully occupy the open space between the known classes"?

---

> ### Author Response · Authors · 2018-11-20
> **Detailed response**
>
> We thank the reviewer for this comment.
>
> We have conducted further experiments to quantify the advantage of  ii-loss over central loss. Here are the results from experiments on using central loss:
> MNIST dataset: 30-run average AUC=0.9264
> Android dataset: 30-run average AUC=0.7514
> MS dataset: 30-run average AUC=0.9234
>
> In all cases ii-loss based approach records a statistically significant improvement over central loss based approach.
>
> The authors of triplet-center loss (which incorporates inter-class distance) also found improvement over center-loss in (regular) classification tasks.   Also, although both ii-loss and triplet-center loss consider inter-class distance, ii-loss has less computational overhead.   In ii-loss, centroids (centers) are first calculated and inter-class distances between centroids are then calculated.  In triplet-center loss, for each sample, the closest sample from another class need to be found.
>
> With regards to the reviewers question about how to differentiate the known class outlier and new class:
> - Our work focuses on differentiating new class rather than on the problem of known class outliers. Although we believe the proposed approach can be applied to the known class outlier problem we have not conducted experiments to verify this.
>
> With regards to heterogeneous unknown classes:
> - The open set problem is actually easier when the unknown classes are more different from the known classes.
>
> The following are answers to the detail questions:
> 1. There shouldn’t be much of a difference between optimizing the ii-loss and cross-entropy loss separately or simultaneously. Our decision to do this separately was many an engineering decision to allow as to experiment with different network topology.
> 2. In our experiments we found the 1% contamination ratio to be widely applicable. We chose contamination ratio based threshold because it is easier to understand by the user. For example, if catching more unknown class instances is more important to the user she/he can set a higher contamination ratio and vise versa.
> 3. When the unknown class and the known class are different (for example the reviewer mentions animals vs scene) the unknown classes are not projected close to any articular known class hence occupying more space between the known classes. The more difficult problem is when the unknown classes are very similar to the known. In which case their projection  will be closer to one of the known and might even overlap making it more difficult. Hence why in the classes in our experiments are similar.
> 4. The motivation for making "the unknown class instances fully occupy the open space between the known classes" is to reduce the chance of unknown class projection overlapping a known class projection.
>
> Finally, we would like to point out that the papers on center loss and triplet-center loss did not  address the task of open-set recognition.  However, our paper focuses on open-set recognition. Also, the papers on center loss and triple-center loss focus on computer vision-related problems.  However, our paper discusses the utility of our proposed approaches in two very different domains: computer security and computer vision.

---

### Official Review · AnonReviewer3 · 2018-11-02
**The paper focuses on open set classification which is a clearly desirable feature for all machine learning classification algorithms. The idea is very simple and the experiments should include more comparative results with the baselines.**

**Rating:** 4
**Confidence:** 4

**Review:**


The paper focuses on open set classification where one wants to design a classifier able to accurately classify samples from training (known classes) and able to reject samples from unknown classes. Such a feature is would be clearly desirable for all machine learning classification algorithms. The paper presents a representation learning based approach for this problem.

The idea is very simple. It consists in learning a neural classifier with a constraint on the representation space of samples (i.e. the one implemented in a chosen hidden layer of the network) aiming at optimizing a Fisher-discriminant-like criterion. This criterion aims at minimizing the variance of the representation of the samples within a class and to make representations of samples from different classes (actually the means per class) well separated. The learning is eventually performed by adding a usual cross entropy classification loss on the output layer to the Fisher like criterion. The rejection of samples from unknown classes is performed via a threshold on the minimum distance of a sample representation and the class means in the representations space.  The idea is well thought but the innovation is indeed low.

Experiments are performed on three, but small, datasets including the simple Mnist dataset. Experimental results compare the proposed approach and its variants to two recent baselines, OpenMax and G_OpenMax. Experiments show the proposed approach outperform the two baselines but in some cases the confidence interval is quite large and prevent definitive conclusions (e.g. up to 0.05 in Table 2). Visualization of projected data show as expected the interesting feature of the representation space. Yet the experimental analysis does not seem as deeps the ones in the two papers where baselines were published. For instance results are shown with respect to a measure of the experimental setting named openness in [Ge and al.]. Moreover the paper by  [Ge and al.] conclude to the superiority of their proposal with respect to OpenMax which is not fully consistent with the results reported in this paper. Also experiments were performed on much bigger datasets in these two references with ILSVRC 2012 dataset in [Bendale and al., 2016] while [Ge and al.]  used a handwritten diet dataset with more than 350 classes. It would drastically strengthen the paper if the authors could provide comparative results on these datasets too.

---

> ### Author Response · Authors · 2018-11-20
> **Difference in experimental results**
>
> We thank the reviewer for this comment.
>
> The inconsistency between results reported in g-openmax and our work can be down to the evaluation methodology. Since the evaluation methodology used by g-openmax is not clearly stated in the g-openmax paper, it difficult to exactly reproduce the results. That being said we have clearly detailed our experiment methodology and use the same experiment  framework for all of the evaluated approaches.

---

### Official Review · AnonReviewer2 · 2018-11-02
**A reasonable  paper needing further justifications**

**Rating:** 4
**Confidence:** 4

**Review:**

The paper proposed a NN-based model for open set recognition via finding a better feature space where larger inter-class (P2) and smaller intra-class distances (P1) are satisfied. In the proposed model, the inter- and intra-class distances are measured basing on the mean of final linear layer features from each class, and a kind of L2 loss is defined to ensure the properties of the feature space during the training progress. Then the proposed outlier score defined as the minimum inter-class distance becomes the key for the open set recognition task.

Generally, this paper is well written and easy to read. The proposed threshold estimation method for outlier score based on assumed contamination ratio is reasonable. And three datasets in two domains are used to prove the model’s effectiveness.

Major comments:
1.	This paper seems less novel.
There exist several methods aiming to find a better feature space satisfying the mentioned feature distance properties by adjusting the optimized loss functions, such as Center Loss (Wen et.al 2016) and Additive Angular Margin (Deng et.al 2018). I think the idea Combining ii-loss with Cross Entropy Loss proposed in this paper is quite similar to the Center Loss except that
a)	the ii-loss contains a part for maximizing the minimum inter-class distance;
b)	the ii-loss and cross entropy loss are optimized separately.
Since results shown in Center Loss that without pushing the inter-class distance, the feature space still satisfied P1 and P2, this paper seems not that novel, at least some comparation can be added to analyze the improvement for adding the inter-class part.

2.	This paper seems less convincing as well.
The paper introduces that the two properties (larger inter-class and smaller intra-class distances) can lead to larger spaces among known classes for the instances of the unknown classes to occupy. However, it keeps uncertain if this can be generalized to unseen classes. In this sense, it is better to conduct some additional theoretical analysis or perform more experiments to validate this. In particular, only one plot was performed to verify this point on one single dataset. Maybe more plots of the distributions can be provided on more additional datasets.

---

> ### Author Response · Authors · 2018-11-20
> **Additional baseline**
>
> We thank the reviewer for this comment.
>
> As the reviewer noted  there are differences between the central loss and our proposed ii-loss
> 1. Our loss function has an additional component for maximizing the minimum inter-class distance.
> 2. The ii-loss and cross entropy loss are optimized separately.
> 3. ii-loss can be used without cross entropy loss
> 4. Our proposed approach shows the applicability of such feature learning to open set recognition problem.
>
> We have conducted further experiments to quantify the advantage of adding the inter-class distance term in ii-loss. In these experiments we use central loss to train:
> MNIST dataset: 30-run average AUC=0.9264
> Android dataset: 30-run average AUC=0.7514
> MS dataset: 30-run average AUC=0.9234
>
> In all experiments ii-loss based approach records a statistically significant improvement over central loss based approach.
>
> Finally, we would like to point out that the paper on center loss did not address the task of open-set recognition.  However, our paper focuses on open-set recognition. Also, the papers on center loss and triple-center loss focus on computer vision-related problems.  However, our paper discusses the utility of our proposed approaches in two very different domains: computer security and computer vision.

---

### Public Comment · ~ioui_wc1 · 2018-10-21
**question on generalisation**

Hi, I have a question on reducing overlap in feature space between samples from train categories and unknown categories.
I agree that maximizing inter-class separation and minimizing the intra-class separation is beneficial and indeed can reduce overlap in the feature space. However, as far as I known from the metric learning field, this benefit is only true when testing on samples from trained categries, and it will not generalise to samples from untrained unknown categories, except that you really get enougth data to trained a generic metric learning network. That is, samples from untrained unknown categories, when mapped into the feature space, will not have desired non-ovelapping ranges. As you can see from Figure 2, the distribution of trained categories is ideal, while the distribution of untrained categories just are a mess, and worsely, overlapping with the distribution of trained categories. so that in fundamental, your idea to maximize inter-class separation and minimize the intra-class separation seems not generalise well to untrained catogories.

---

> ### Author Response · Authors · 2018-11-20
> **clarification**
>
> Than you for your comment.
> Figure 2 is shows two figures z0-z1 and z0-z2 , where z0, z1, z2 are there dimensions from the size dimensional z-layer. Overlap in these figures does not necessary mean real over lap. It simply means it appears as over lap when viewed from that dimensions perspective. This is backed by the improved open set recognition performance shown in the experimental section.

---

### Meta-Review · Area_Chair1 · 2018-12-14

**Confidence:** 4
**Recommendation:** Reject

**Metareview:**

The paper presents an approach to address the open-set recognition task based
on inter and intra class distances. All reviewers are concerned with novelty
and more experimental comparisons. Authors have added some results, but
reviewers did not think these were enough to make the paper convincing enough.
Overall I agree with reviewers and recommend to reject the paper.